# Comment on Manna et al. SARS-CoV-2 Inactivation in Aerosol by Means of Radiated Microwaves. *Viruses* 2023, *15*, 1443

**DOI:** 10.3390/v15102110

**Published:** 2023-10-18

**Authors:** Gavin J. Taylor, Jérémie Margueritat, Lucien Saviot

**Affiliations:** 1Institute for Globally Distributed Open Research and Education (IGDORE), São Carlos 13560-230, Brazil; 2Institut Lumière Matière, UMR5306, Université de Lyon, 69622 Villeurbanne, France; jeremie.margueritat@univ-lyon1.fr; 3Laboratoire Interdisciplinaire Carnot de Bourgogne, UMR 6303 CNRS–Université de Bourgogne Franche Comté, 21000 Dijon, France; lucien.saviot@u-bourgogne.fr

In a recent article published in *Viruses* by Manna et al. [1], the authors estimate the resonant frequency of SARS-CoV-2 particles using elastic continuum theory of the vibrational modes of a sphere [2]. Unfortunately, the article provides an incorrect solution to the eigenvalue equation. Although this does not influence the validity of the experimental results, the error will be misleading if the solution is used to select frequencies for testing virus inactivation in other studies. This comment provides details on how to obtain a correct solution numerically.

Elastic continuum theory can be used to calculate the resonant frequencies of the vibrational modes of a homogenous isotropic free sphere [2]. This is often simply referred to as Lamb’s theory or equation, and only requires the radius of the sphere, as well as its longitudinal and transverse sound velocities, as input (Appendix A). Recent studies have investigated microwave resonant absorption as a mechanism for inactivating spherical viruses [3,4,5,6,7,8]. These studies propose that virus inactivation results from a structural-resonant energy transfer (SRET) process, where electromagnetic fields matching the frequency of the dipolar mode of a virus capsid excite a confined acoustic vibration which damages its structure [9]. Although most inactivation studies have empirically tested the effectiveness of a range of microwave frequencies, the frequency of the dipolar mode can be measured directly using microwave absorption spectroscopy [9,10,11,12,13,14]. The peak absorption frequency from spectroscopy can then be used as the centre of a frequency range to test for inactivation experiments [7,9,10]. Alternatively, Lamb’s equation can be used to calculate the dipolar resonant frequency across the size range of the virus, which indicates a frequency range over which inactivation can be tested [1,15]. However, while the dimensions of many viruses have been measured using electron microscopy, acoustic properties are only available for a single virus (STMV [16]). The applicability of Lamb’s equation to spherical viruses is also limited, as they are inhomogeneous, many are only approximately spherical, and experiments are usually performed with viruses suspended in a viscous liquid [17]. Nonetheless, Lamb’s theory can still provide a useful estimate of the dipolar resonance frequency for designing or analysing inactivation experiments.

To estimate the resonant frequency (*f*) of SARS-CoV-2, Manna et al. [1] derived an analytical solution to Lamb’s eigenvalue equation for the first harmonic of the dipolar mode of a sphere (their Equation (1)). Their solution starts by assuming that the ratio between the longitudinal and transverse sound velocities is 2, and uses an approximate expression for the asymptotic value of a Bessel function when its argument approaches zero (their Equation (2)). Manna et al. [1] used this approximation to replace the spherical Bessel functions in the eigenvalue equation with polynomial functions of the eigenvalues *ξ* and *η* (for completeness, we note that the approximation for small arguments of a spherical Bessel functions is: $j_{m}\left(z\right)\approx \frac{z^{m}}{\left(2m+1\right)‼}$ [18] (p. 708), while their work uses the approximation for cylindrical Bessel functions). However, after performing this substitution, they omitted the multiplication by *ξ* (which is present in their Equation (1)) from the first term of their Equation (3). Consequently, their Equation (4) should be corrected to *η*^2^ = 2*ξ*^2^. Substituting *η* = 2*ξ* into the corrected Equation (4) then leads to *ξ* = 0, which produces a result of *f* = 0 regardless of the radius or sound velocity. The same error has also been made elsewhere [15]. Although trivial, the 0 Hz solution is, in fact, correct, and corresponds to translations of the sphere. However, this approach does not indicate the resonant frequency of the dipolar mode, which is the desired result for selecting the frequency range to test in microwave inactivation experiments. A better approach is to solve the eigenvalue equation numerically (Appendix A).

We provide a python script which numerically calculates the solution for the frequencies of the vibrational modes of a sphere (github.com/gavinscode/sphere_resonance [19] (accessed on 5 August 2023)). Additionally, a web app (saviot.cnrs.fr/lamb, [20] (accessed on 5 August 2023)) can be used to calculate the frequencies and visualise the corresponding displacements of the resonant modes. Both of these tools can be used regardless of the ratio between longitudinal and transverse sound velocities, and they can also calculate the frequencies of all the spheroidal resonant modes and harmonics (Appendix A).

Following Manna et al. [1], we calculated the resonant frequencies of virus particles from 30 to 70 nm in radius (60 to 140 nm in diameter [21]), with longitudinal and transverse sound velocities of 1800 m/s and 900 m/s, and estimated that the resonant frequency of SARS-CoV-2 is between 7.4 to 17.2 GHz. Using Equation (5) from Manna et al. [1] with the parameters above provided a frequency range of 12.9 to 30 GHz, although the initial estimate of 6.5 to 16 GHz seems to be based on mistakenly using the virus diameter in place of its radius in Equation (5). In their preliminary experiments, Manna et al. [1] tested inactivation of SARS-CoV-2 with frequencies from 6.5 to 17 GHz, and found that inactivation peaked at 10 GHz. The SRET paradigm suggests that this inactivation peak corresponds to the dipolar resonant frequency of the virus, although thermal mechanisms may have contributed to the observed inactivation, and additional analysis is required in order to exclude these [3]. However, Manna et al. [1] would not have found this inactivation peak if they had only tested frequencies across the range of 12.9 to 30 GHz resulting from their Equation (5), whereas our solution does indicate a range (7.4 to 17.2 GHz) that includes the 10 GHz peak. As such, we recommend that future research on virus inactivation use a numerical solution for Lamb’s equation when an estimate of the dipolar resonant frequency is required.

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
