# Peer review of "Comment on Manna et al. SARS-CoV-2 Inactivation in Aerosol by Means of Radiated Microwaves. Viruses 2023, 15, 1443"

_viruses, 2023, doi:10.3390/v15102110_

Round 1

Reviewer 1 Report

The manuscript viruses-2577322 entitled Comment on Manna et al. SARS-CoV-2 Inactivation in Aerosol by Means of Radiated Microwaves. Viruses 2023, 15, 1443 by Gavin J. Taylor and coworkers, evidenced that in a recent article published in Viruses by Manna et al. [1], the authors estimate the resonant frequency of SARS-CoV-2 particles using elastic continuum theory of the vibrational modes of a sphere. The article provides an incorrect solution to the eigenvalue equation that will be misleading if used to estimate the resonant frequency of viruses in other studies.

The work was appropriate, the equation properly resolved.

The discussion is consistent with the work

English lenguage is appropriate

Author Response

We thank the reviewer for taking the time to read our article and expressing their interest in the work. 

Reviewer 2 Report

This article is interesting. Congratulations for it but for me due to my speciality it was hard to evaluate it.

Moderate english revision

Author Response

(The authors gave the same response as above.)

Reviewer 3 Report

This manuscript is a correction for the work that was conducted by Manna et al. The authors of this manuscript have identified that Manna et al. used an incorrect solution, which could lead to an inaccurate calculation of the resonant frequency of the virus.

I have reviewed the correction manuscript titled "Correction Manuscript for Resonant Frequency Calculation in Manna et al." I commend the efforts of Taylor et al. in addressing the error identified in the original article and providing accurate calculations for the resonant frequency. The correction manuscript is well-organized and clearly outlines the nature of the error, the methodology used for recalculations, and the revised resonant frequency values. However, there are some concerns that I would like to address:

  1. It would be advisable to consult with the authors of Manna et al. to obtain their comments on Taylor’s findings. The calculations performed by Taylor et al. may not be comprehensive, and certain factors mentioned in the original manuscript might not have been considered.
  2. On page 2, Taylor et al. mention that "SARS-CoV-2 has particles ranging from 30 to 70 nm in radius." This statement is not accurate. In reality, the virus size falls within the range of 70 to 100 nm. If Taylor et al. included these data in their calculations, it might result in an incorrect resonant frequency calculation.
  3. The manuscript by Manna et al. presents both computational and experimental results. To elaborate, Manna et al. discussed the inactivation ratio of SARS-CoV-2 when exposed to illuminating microwaves at frequencies ranging from 6.5 to 17 GHz. Manna concluded that viral inactivation at 10 GHz could exceed 60%. This experimental outcome aligns with their calculations. Therefore, any potential miscalculation may not significantly impact their final conclusions about the resonant frequency.
  4. We appreciate Taylor et al.'s effort to inquire whether the incorrect solution could lead to miscalculations and incorrect conclusions regarding SRET-mediated viral inactivation. This topic is highly controversial, and a precise analysis of SRET-mediated inactivation while excluding unwanted thermal effects is crucial.

Best regards,

Author Response

We thank the reviewer for taking the time to read our article and expressing their positive comments on the work. 

Regarding the reviewer’s concerns:

  1. We agree that allowing Manna et al. an opportunity to reply is warranted. However, we clearly demonstrate the solution they provide for the dipolar mode of Lamb’s equation is incorrect.  Although this does not substantially influence the conclusions of their paper, we wrote this comment as we were worried that the error could mislead other researchers in the field, some of whom may not have used Lamb’s theory before. As indicated in our comment, the mistake Manna et al. made is also present in another recent article (Barbora, A.; Minnes, R. Targeted Antiviral Treatment Using Non-Ionizing Radiation Therapy for SARS-CoV-2 and Viral Pandemics Preparedness: Technique, Methods and Practical Notes for Clinical Application. PLOS ONE 2021). We hope that our timely and detailed comment will ensure that future researchers use the theory correctly. 
  2. We used the same reference for the virus size as Manna et al. (Scheller, C.; Krebs, F.; Minkner, R.; Astner, I.; Gil-Moles, M.; Wätzig, H. Physicochemical properties of SARS-CoV-2 for drug targeting, virus inactivation and attenuation, vaccine formulation and quality control. Electrophoresis 2020) which indicates “SARS-CoV-2 … the diameter varying between approximately 60 to 140 nm”. Other articles provide values in this range. We suspect that confusion occurred as we indicated the radius (30 to 70 nm), whereas diameter is a more common measurement of virus size. Incidentally, Manna et al. also appear to have mistakenly used the SARS-COV-2 diameter as its radius (see below). To avoid confusion, we add the diameter to the text.
  3. We agree that the 10 GHz inactivation peak in their experimental results is clear, and have added a sentence in the discussion paragraph noting that this is likely to correspond to the dipolar resonant frequency, as well as a sentence in the abstract that their error does not influence the conclusions of the experimental work. However, our main concern is that by using the incorrect solution in Manna et al. other investigators may select a frequency range to test that does not cover the dipolar resonant frequency. The initial frequency range estimate (6.5-16 GHz) provided by Manna et al. does cover the dipolar resonance, but they appear to have arrived at this estimate by mistakenly using the virus diameter (60-140 nm) in place of radius in their Equation 5. If they had used the radius, Equation 5 would have provided 12.9-30 GHz, and they may not have tested frequencies below 10 GHz. Conversely, correctly using the radius with our solution does lead to an estimate (7.4-17.2 GHz) that includes the resonance. We have tried to emphasise the consequences of making this error for experimental design in the final paragraph. 
  4. We agree that additional analysis of SRET-mediated virus inactivation is warranted and have added a note in the final paragraph that an investigation that excludes thermal effects is required before the 10 GHz peak can be confidently attributed to SRET.

Round 2

Reviewer 3 Report

I am pleased to acknowledge that the authors have diligently addressed the issues raised in the comments on the Manna et al. article.

The primary concerns pertained to the virus size, the frequency range selected for testing, and potential unwanted thermal effects.

However, one concern still remains: prior to finalizing the publication of this comment article, it is advisable to ensure that the authors of the original articles are informed about the comments and given the opportunity to provide their response. Notably, this comment article pertains to two distinct articles, namely the Manna et al. article and the Barbora et al. article. It is crucial to obtain their input and response prior to finalizing the publication of this comment article.